# An Approach to the Diversity of Achelata and Brachyura (Crustacea, Decapoda) from the Southern Mexican Caribbean

Jani Jarquín-González [1], Martha Valdez-Moreno [2,*] and Rigoberto Rosas-Luis [1,3,*]

1 Tecnológico Nacional de México/I.T. de Chetumal, Av. Insurgentes 330, Chetumal 77013, Quintana Roo, Mexico
2 El Colegio de la Frontera Sur, Av. Centenario km 5.5, Chetumal 77014, Quintana Roo, Mexico
3 CONACyT-Tecnológico Nacional de México/I.T. de Chetumal, Av. Insurgentes 330, Chetumal 77013, Quintana Roo, Mexico
* Correspondence: mvaldez@ecosur.mx (M.V.-M.); rigoberto.rl@chetumal.tecnm.mx (R.R.-L.)

**Abstract:** Decapods include species of economic importance, such as Achelata (lobsters) and Brachyura (true crabs), since they have aesthetic, commercial, gastronomic, and biomedical value. These groups exhibit a great variety of shapes, larval stages, habits, and sizes, making them difficult to recognize. In the Southern Mexican Caribbean (SMC), no taxonomic list or analysis of the biological diversity for the Achelata and Brachyura has been performed. Herein, the biological diversity of these groups was analyzed by reviewing the literature and collecting specimens in the SMC to obtain morphological, ecological, and molecular data. These results showed a total of 29 families, 67 genera, and 98 species recorded, of which, one is considered as a potentially new species, six are new records for the SMC, 12 expanded their distribution range, and 14 species names were updated. In addition, the BOLD system assigned 21 BINs supported with morphological identification. This work contributes positively to the knowledge of the marine and coastal decapods from the SCM as it represents the first effort to recognize their current biological diversity. This information will be used to develop adequate strategies for the conservation and management of marine and coastal natural resources of the SMC.

**Keywords:** crustaceans; biodiversity; barcodes; new records; Atlantic

## 1. Introduction

The Order Decapoda belongs to the class Malacostraca (subphyla: Crustacea). It includes a wide variety of organisms: the true crabs (Brachyura), hermit and porcelanid crabs (Anomura), shrimps (Dendrobranchiata, Caridea, and Stenopodidea), and lobsters (Astacidae, Achelata) [1,2]. Decapods include species of economic importance, penaeid shrimps, palinurids, lobsters, portunids, and crabs since they have aesthetic, commercial, gastronomic, and biomedical value [3–5]. However, the most relevant decapods belong to the infraorders Achelata and Brachyura as they support some of the most remunerative fisheries worldwide [4,6].

These crustaceans live in marine and coastal ecosystems and fulfill different ecological functions. They are relevant in marine, pelagic, and benthic trophic networks due they serve as food for birds, marine mammals, sharks, turtles, starfish, cephalopods, and fish of commercial importance [7–9]. It has been documented that more than 50% of the diet of adult and juvenile snappers is based mainly on decapod crustaceans [9,10]. They also regulate the herbivore populations (e.g., sea urchins), favoring primary and secondary production and the trophic structure's stability [7]. Furthermore, when some species of decapods (e.g., *Rhithropanopeus harrisii*) are introduced or invade a new region successfully, they become exotic species, and they can modify the native marine communities by altering habitat and ecosystem function [11,12].

Despite their economic and ecological importance, decapod species show severe taxonomic difficulties due to their high variety of shapes, larval stages, habits, and sizes. Martin

& Davis [3] published an updated classification of the Crustacea where they highlighted the necessity for reaching a consensus on the relationships among the Decapoda because opinions and datasets remain sharply divided. According to Álvarez et al. [2], in Mexico, the species richness of decapods is about 1775 species classified in 537 genera and 115 families, representing 11.9% of the total species and 57.5% of the families in the world, respectively. Also, of the total number of these species, 1597 (89.9%) are marine, and 178 (10.1%) are freshwater; 46.7% of the marine species occur in the Mexican Pacific, 31.4% in the Gulf of Mexico, and 21.8% in the Mexican Caribbean.

In the Mexican Caribbean, some studies have been conducted to know the biological diversity of marine and coastal crustaceans, including the infraorders Achelata and Brachyura. Markham et al. [13], García-Madrigal et al. [14], and Álvarez [15] reported species lists for different orders of crustaceans (e.g., Stomatopoda, Peracarida, Decapoda) from the shallow Caribbean coast of Quintana Roo between 1990 and early 2000, since then, the information has not been updated. In the north coast of the Mexican Caribbean (Isla Mujeres and Puerto Morelos, Quintana Roo), Campos-Vázquez [16], González-Gómez et al. [17] and Briones-Fourzán et al. [18], conducted studies of decapods associated to coralline reefs and seagrass, respectively. In contrast, the Southern Mexican Caribbean has not been studied, and there is no taxonomic list of species or an analysis of the current biological diversity for the decapods Achelata and Brachyura.

The analysis of the biological diversity and the taxonomic and genetic status of different groups has been performed using molecular sequence data as the DNA barcodes [1,19–23]. This analysis mainly used mitochondrial gene cytochrome c oxidase subunit I (COI), providing a robust species-level resolution for different groups of animals, such as marine decapods [19,22,24]. As a result of the use of DNA barcodes and the inclusion of morphology description, Costa et al. [19] and Landschoff & Gouws [21] proposed the possible arbitrary threshold for genetic species delimitation in this group can be placed between 3.7% and 4.9%. In addition, DNA barcodes have proven to be a helpful tool in species differentiation in the last two decades, accelerating biodiversity inventories [24] to assign unknown specimens to already described and classified species, improving the discovery of new species, and facilitating their identification, particularly in cryptic, microscopic, and other organisms with complex morphology [25]. The Barcode of Life Data System (BOLD) is "an informatics workbench aiding the acquisition, storage, analysis and publication of DNA barcode records" [26]. This system allows the exchange of genetic and biological information and scientific collaborations. In addition, the genetic data can be associated with photographs, collection site metadata, life stage, and samples with vouchers in scientific collections [24]. Also, can assign a Barcode Index Number (BIN), equivalent to a Molecular Operating Taxonomic Unit (MOTU) for all samples that cover minimum information standards; with this, a standardized reference is made for unidentified organisms [27].

Assessing the current biological diversity correctly [28,29] and the ecological role of decapods in the SMC are priorities in understanding the geographic distribution patterns, taxonomic, genetic, and conservation status of these species. This knowledge is essential to make sustainable use of natural resources since there are currently various threats to public health, ecosystem health, fishing, and the economic development of the region, such as the growing arrival of *Sargassum* in the Mexican Caribbean, which generates hypoxia and deterioration of water quality, affecting individuals of a large number of species, mainly fish and crustaceans [30]. Therefore, the main objective of this study was to know the current state of the biological diversity of the Brachyura and Achelata decapods present in the SMC. The results will allow the development of instruments and strategies in favor of environmental sustainability and the conservation of the natural capital of the Southern Mexican Caribbean ecosystems.

## 2. Materials and Methods

### 2.1. Literature Review

The reports of Brachyura and Achelata species recorded in the Southern Mexican Caribbean (SMC) were analyzed to elaborate a taxonomic list. The Scopus, Google Academic, and Springer Link databases were consulted from October 2020 to November 2021. The specific search terms for revision were Decapoda, Brachyura, and Achelata (including the addition of the words COI, marine, native, exotic, and introduced), Mexican Caribbean, and the Atlantic Ocean. Additionally, to search the species records in the study area, the databases of the Reference Collection of Benthos of El Colegio de la Frontera Sur (ECOSUR), Chetumal, Mexico, and the online dataset of the National Collection of Crustaceans of Universidad Nacional Autónoma de Mexico (UNIBIO [31]), were visited and consulted. The online project "Stomatopod, amphipod, isopod and decapod crustaceans of the Quintana Roo coast" by Álvarez [15] was also included in the taxonomic revision. For each collection database, the search was restricted to localities from Punta Herrero (19.31157, −87.44517) to Xcalak (18.27001, −87.82649), Quintana Roo, Mexico. The BOLD Public Data Portal [32] (www.boldsystems.org, accessed on 13 July 2022), was consulted to obtain and compare molecular data. The taxonomic list includes the name of species, type locality, distribution, ecological notes, the process ID for the sequences obtained in this work, and BINs.

### 2.2. Material Collected

Punta Herrero, Punta Diamante, El Uvero, Mahahual, Bermejo River, North of Xahuayxol, Xahuayxol, and Huach River, in the Southern Mexican Caribbean were selected to collect individuals during 2021: January (27th to 29th), February (22nd to 24th), April 2021 (17th to 19th), October (27th to 29th), and November (10th to 12th) (Figure 1). At each site, samples of decapods were collected manually during free diving samplings, mainly of coralline rocks associated with algae at depths not exceeding 3 m (collection permit PPF/DGOPA-060/20). The sampling effort was 5 h by site and was performed by two persons. Individuals were photographed in situ with a digital camera (Sony DSC-H300) and fixed in 96% ethanol. Large specimens (more than 3 cm of carapace length) were injected with the same ethanol in the articulation between segments of the body. Subsequently, the samples were stored at −20 °C for a week to prevent DNA degradation [33]. The collected material was examined under a stereoscopic microscope Zeiss StemiDV4 and identified according to Rathbun [34–36], Williams [37–39], and Abele & Kim [40]. All material analyzed was then incorporated into the Reference Collection of Benthos (ECOSUR), Chetumal, Mexico. The species list was arranged alphabetically, and the nomenclatural status of each specie was assigned following the WoRMS Editorial Board [41].

### 2.3. Molecular Analysis

For molecular analysis, 76 specimens were processed. A small piece of muscle (1–3 mm$^3$) or 2–3 eggs (in the case of ovigerous females) were used to extract the DNA. The forceps and the material were sterilized using a solution of 1:5 chlorine/distiller water and subsequently neutralized with ethanol 96% between each tissue extraction.

A lysis buffer was used to digest each sample's tissue with proteinase K, all samples were digested in an oven for 12 h. at 56 °C. The extraction was carried out through a 1.0 mm PALL glass fiber plate [42]. Cytochrome Oxidase I (COI) gene segment with an approximate length of 650 Bp [43] was amplified using the zooplankton primers [44]. Amplification was carried out with a final volume of 12.5 µL, prepared as follows: 6.5 µL of 10% trehalose, 2 µL of ultrapure water, 1.25 µL PCR buffer X10, 0.625 µL MgCl$_2$ (50 mM), 0.125 µL of each Primer (0.01 mM), 0.06525 µL dNTP mix (10 mM), 0.625 µL Taq polymerase, and 2 µL of DNA template. The reactions were cycled at 94 °C for 1 min, followed by five cycles at 94 °C for 30 s, 45–50 °C for 40 s, and 72 °C for 1 min, followed by 35 cycles at 94 °C for 30 s, 51–54 °C for 40 s and 72 °C for 1 min, finally one last cycle of 72 °C for 10 min.

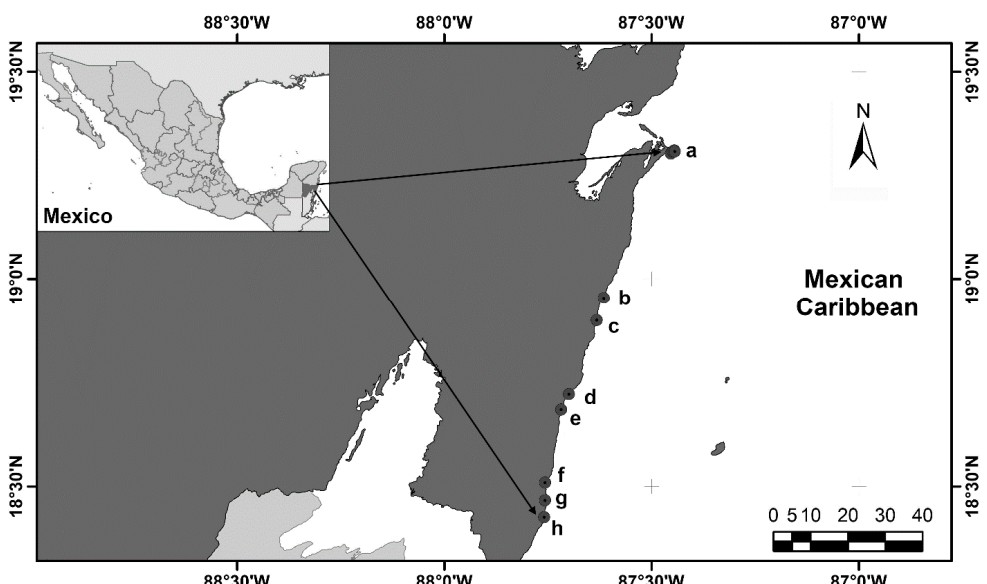

**Figure 1.** Localities within the Southern Mexican Caribbean where decapods were collected. (a) Punta Herrero; (b) Punta Diamante; (c) El Uvero; (d) Mahahual; (e) Bermejo River; (f) North of Xahuayxol; (g) Xahuayxol; (h) Huach River. Figure edited by Jani Jarquín-González.

The PCR products were visualized in agarose gel *Invitrogen^TM* with four μL of sample and 16 μL of water. The PCR products were sequenced at Macrogen (Seoul, Republic of Korea). Finally, the sequences were edited with Codon code v.3.0.1 and uploaded to BOLD. Specimen images, field data, and COI sequences obtained in this study can be consulted at BOLDSYSTEMS www.boldsystems.org within the dataset ID: DS-DECA01 Decapoda (Achelata and Brachyura) from Southern Mexican Caribbean (https://doi.org/10.5883/DS-DECA01, accessed on 13 July 2022).

All the sequences obtained were compared with COI sequences previously published using the specimen identification tool in the Barcode of Life Data System (BOLD) [45]. Similarity values >98% were considered for all identified species, which confirmed their placement under different numbers of BINs [26,28]. The Kimura 2-parameter model (K2P) was used to calculate the genetic divergences between the species [46] and the maximum likelihood (ML) tree. The BOLD ID tree was simplified using the compression feature provided by MEGA X software [47]. The criteria to assign taxonomic level identification using BOLD was a similarity value $\geq$ 99% [45].

## 3. Results and Discussion

### 3.1. Literature Review

Results of the literature review showed that 91 species of decapods have been reported in the Southern Mexican Caribbean (Table S1). Of these, 78 species (86%) have type locality within Western Atlantic (e.g., Antilles, USA, Brazil); seven (8%) have type locality outside Western Atlantic (e.g., Africa, Australia, Indonesia, Chile, and the Mediterranean Sea); five (5%) with unknown type locality (e.g., *Panulirus argus* (Latreille, 1804) species of great economic importance); and one (1%) has type locality from the Mexican Caribbean (*Parapinnixa bouvieri* Rathbun, 1918 from Cape Catoche).

Regarding the habitat, 82% of the species were present in marine environments, including reefs, tide pools, rocky or sandy beaches, estuaries (15%), salt marshes (2%), and mudflats (1%) in a lesser proportion. The most important habitats reported for these species were coralline reefs (22%), sediments (sand, gravel, mud) (19%), and rocky bottoms (13%), followed by mollusk shells (10%), sponges (9%), seagrass (8%), algae (8%), other invertebrates (cnidarians, polychaetes, barnacles, echinoderms, foraminifera, and tunicates) (6%), mangrove roots (4%), and *Sargassum* (1%). Additionally, the species *Calappa ocellata*

Holthuis, 1958, *Euryplax nitida* Stimpson, 1859, and *Percnon gibbesi* (H. Milne Edwards, 1853) was reported in stomach contents of bothids, holocentrids, diodontids, labrids, lutjanids, and serranids (Table S1).

Results of the number of records by locality showed that the localities with the highest number of recorded species were Mahahual (20%), Xahuayxol (17%), and El Placer (13%), followed by Banco Chinchorro and El Uvero (10%), Rio Indio (9%) and Chetumal Bay (5%). In contrast, the localities with the lowest number of records were Punta Herradura (4%), Xcalak (4%), and Huach River (3%). Of all species, 60% of species were recognized only in one specific location, the remaining 40% were recorded in more than one locality. Additionally, herein 14 names of the species found in the literature were updated (Table 1).

**Table 1.** Update of the species names of Brachyura and Achelata found in the bibliographic review. SMC = Southern Mexican Caribbean.

| Old Name | References of This Record in the SMC | Current Name |
|---|---|---|
| *Callinectes larvatus* | [13,14] | *Callinectes marginatus* |
| *Cronius tumidulus* | [14] | *Achelous tumidulus* |
| *Dromidia antillensis* | [16] | *Moreiradromia antillensis* |
| *Micropanope nuttingi* | [13,14] | *Scopolius nuttingi* |
| *Microphrys antillensis* | [13,14,16] | *Omalacantha antillensis* |
| *Microphrys bicornutus* | [13,14,16] | *Omalacantha bicornuta* |
| *Mithrax coryphe* | [13,14] | *Mithraculus coryphe* |
| *Mithrax forceps* | [13,14,16] | *Mithraculus forceps* |
| *Mithrax sculptus* | [13,14] | *Mithraculus sculptus* |
| *Paractaea rufopunctata nodosa* | [14] | *Paractaea nodosa* |
| *Pinnixa floridana* | [14] | *Glassella floridana* |
| *Podochela riisei* | [13,14] | *Coryrhynchus riisei* |
| *Portunus ordwayi* | [13,14] | *Achelous ordwayi* |
| *Xanthodius denticulatus* | [13,14] | *Williamstimpsonia denticulatus* |

### 3.2. Morphological Identification of Specimens Collected in the SMC

A total of 102 specimens were collected in the SMC. According to the morphology, they corresponded to 21 morphotypes, of which 20 were identified at the species level and one assigned to the genus *Panopeus* (Table 2). Six species represent new records for the SMC: *Maguimithrax spinosissimus* (Lamarck, 1818), *Menippe nodifrons* Stimpson, 1859, *Mithraculus cinctimanus* Stimpson, 1860, *Mithrax tortugae* Rathbun (1920), *Plagusia immaculata* Lamarck, 1818, and *Portunus sayi* (Gibbes, 1850).

Considering the 91 records found in the literature review and the inclusion of the six new records and the morphotype, the result is a total of 98 species of decapods for the infraorders Achelata and Brachyura from the Southern Mexican Caribbean. Of these, the Achelata included two families, three genera, and four species; and Brachyura has 27 families, 64 genera, and 94 species (Table S1). Also, of the 98 species recorded only 2% (*P. bouvieri* and *Panopeus* sp.) are native to the region, while the rest of the species are shared with other places.

**Table 2.** Taxa of decapods (Achelata and Brachyura) recorded in this work and localities where they were found.

| Infraorder | Family | Species | Locality (ies) |
|---|---|---|---|
| Achelata | Palinuridae | *Panulirus argus* | Punta Herrero, Mahahual |
| | Scyllaridae | *Scyllarides aequinoctialis* | Punta Herrero |
| Brachyura | Grapsidae | *Pachygrapsus transversus* | El Uvero, Mahahual, Punta Herrero |
| | Menippidae | *Menippe nodifrons* | Punta Herrero, El Uvero |
| | | *Maguimithrax spinosissimus* | North of Xahuayxol, Mahahual |
| | | *Mithraculus cinctimanus* | South Xahuayxol |
| | | *Mithraculus coryphe* | South of Xahuayxol, Xahuayxol, Punta Herrero |
| | Mithracidae | *Mithraculus sculptus* | El Uvero, Bermejo River, Punta Diamante, Punta Herrero, Huach River, South of Xahuayxol |
| | | *Mithrax pleuracanthus* | El Uvero |
| | | *Mithrax tortugae* | North of Xahuayxol |
| | | *Omalacantha bicornuta* | El Uvero, Bermejo River, Mahahual, Punta Diamante, Punta Herrero, North of Xahuayxol |
| | | *Pitho lherminieri* | South of Xahuayxol, Mahahual |
| | Panopeidae | *Panopeus harttii* | Punta Diamante |
| | | *Panopeus* sp. | El Uvero |
| | Plagusiidae | *Plagusia immaculata* | El Uvero |
| | Portunidae | *Callinectes marginatus* | Huach River |
| | | *Callinectes ornatus* | Punta Herrero |
| | | *Callinectes sapidus* | Punta Herrero |
| | | *Portunus sayi* | Huach River |
| | Pseudorhombilidae | *Scopolius nuttingi* | Punta Diamante |
| | Xanthidae | *Cataleptodius floridanus* | Punta Herrero, Bermejo River, Mahahual, south of Xahuayxol, and Huach River |

According to Álvarez et al. [2], the Mexican Caribbean ranks third place in the biological diversity of marine decapods compared to the Mexican Pacific and the Gulf of Mexico. This low decapod diversity can be explained by few taxonomic studies performed in the region, the sampling of decapods has not been intense and continuous, and the collecting methodologies have been different between the studies carried out [2]. For this reason, it is recommended to increase the sampling in the SMC since, as indicated by Briones et al. [18] and Vargas-Castillo & Vargas-Zamora [47], increasing the sampling frequency during different times of the year, the number of collecting sites, and the different types of substrates are excellent strategies to know the biological diversity of the decapods.

*3.3. Identified Species and Sampled Sites*

After reviewing the faunal composition of each sampled site, it was found that the locality with the highest number of species was Punta Herrero with 10 species (23%), followed by El Uvero with seven species (16%), Mahahual with six species (14%), south of Xahuayxol with five species (12%), Punta Diamante, Huach River, and north of Xahuayxol with four species (9%) each, and Bermejo River with three species (7%). With respect to species frequency, *Mithraculus sculptus* (Lamarck, 1818) and *Omalacantha bicornuta* (Latreille, 1825) were the most predominant species in six of the eight sampled sites; followed by *Cataleptodius floridanus* (Gibbes, 1850) found in five sites; *Mithraculus coryphe* (Herbst, 1801), and *Pachygrapsus transversus* (Gibbes, 1850) in three places; *M. nodrifrons, Panulirus argus* Latreille, 1804, and *Pitho lherminieri* (Desbonne, 1867) in two locations; and the rest of the species were recognized once (Table 2).

Punta Herrero was the locality with the highest number of species because unlike the central zone of the SMC (e. g., Mahahual), it has not experienced important urban, tourist, and economic development [48]. Personal observations of the authors demonstrated that this locality showed a low accumulation of *Sargassum*, contrasting with the impact that the high concentration of *Sargassum* is creating between Mahahual and Xcalak localities. As

the massive influx of *Sargassum* can influence the transformation of the ecosystems and fauna composition of the region [17,18,49], the abundance and diversity of crustaceans will be modified. According to Vargas-Castillo & Vargas-Zamora [49], the species lists are the first step for evaluating temporal changes in the composition and abundance of decapods due to coastal development, pollution, and climate change. To date, there are 15 sites where Achetala and Brachyura crustaceans have been collected and reported for the SMC (including localities from the literature and those studied in this work). These sites are found between Punta Herrero and Xcalak. This area corresponds to approximately a quarter of the total extension of the Mexican Caribbean. Therefore, updating the lists of decapod species, inclusion of other localities, and continuous monitoring of the populations must be priorities for the SMC.

*3.4. Molecular Analysis*

From 21 morphotypes identified, between one and five specimens were selected for molecular analysis. In total, 76 samples were processed and 65 (86%) were amplified correctly. No insertions, deletions, or stop codons were observed in the sequences and the lengths ranged between 608 and 665 base pairs (bp). The average K2P distance between barcode sequences within species was 0.38%, whereas interspecific divergences were 15.18%. These values are within the range reported by Raupach et al. [25] for crustaceans, including the decapod group from the North Sea.

The 65 sequences matched with sequences in the BOLD library with similarity values ≥99%. Based on these results, 10 families with 15 genera and 20 species were identified, and only one specimen was identified to genus level (Figures 2 and S1; Table S1). The BOLD system assigned 21 BINs to the sequences (see project DS-DECA01 in www.boldsystems.org). This result is consistent with the morphological identification of the 20 species and one morphotype (*Panopeus* sp.).

The genus *Panopeus* has 16 species reported in the literature, three for the Eastern Pacific (*P. chilensis*, *P. convexus*, *P. diversus*), one from the Eastern Atlantic (*P. africanus*), and 12 for the Western Atlantic (*P. americanus*, *P. austrobesus*, *P. boekei*, *P. harttii*, *P. herbstii*, *P. lacustris*, *P. meridionalis*, *P. obesus*, *P. occidentalis*, *P. purpureus*, *P. rugosus*, *P. simpsoni*). In the Bold database, there were 12 species with sequences (except for *P. boekei*, *P. convexus*, *P. diversus*, and *P. occidentalis*). The comparison of the sequence of the specimen of *Panopeus* sp., with the genetic material available in BOLD, showed that it did not match any of the species. Regarding the morphology, *Panopeus* sp. is similar to *P. obesus* since both have rounded lateral teeth in the carapace and the distribution of the dark color on the palm of the fixed finger. However, they differ in the body, *Panopeus* sp. showed a brown-white coloring pattern and a marked granular pattern on the carapace, while in *P. obesus* the color of the body is dark purple to russet and has few or no granular patterns on the carapace. Thoma et al. [50], settled that the traditional morphological characters used to assign members of the genus *Panopeus*, have not proven useful, and additional studies (including morphological and genetic data) are necessary to clarify the taxonomic status of the species and their evolutionary relationships. Thus, the authors will continue working with the DNA sequencing of the *Panopeus* sp. found in this work to contrast with morphological characters and assign the correct species or propose a new one for the SMC.

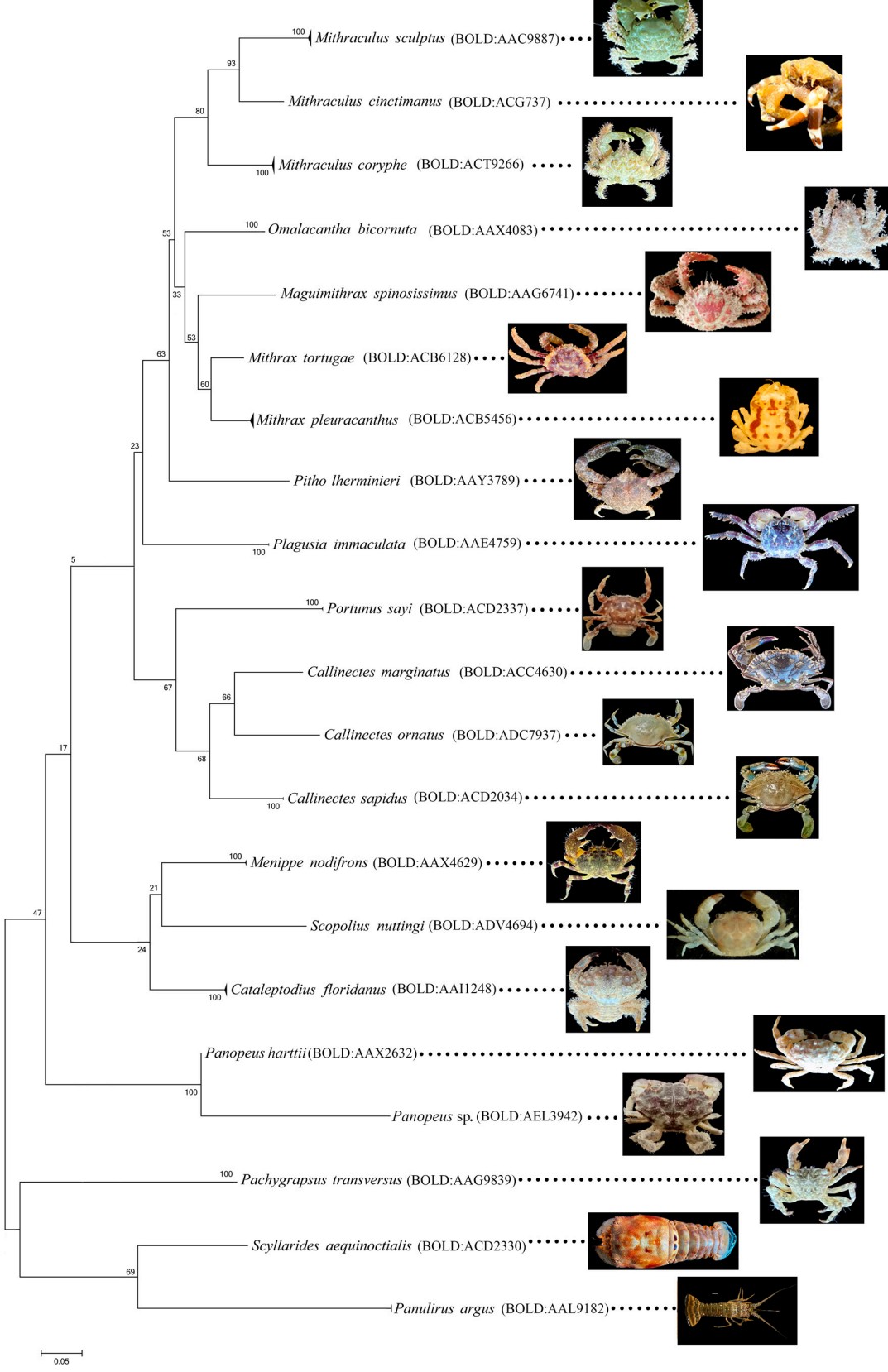

**Figure 2.** ID tree showing the clustering of the 21 morphotypes. The numbers on the branches are the bootstrap support after 500 replicates. Numbers after the names are the BINs.

## 4. Conclusions

The revision of the Achelata and Brachyura species and the inclusion of one potential new species and six new records found in this work resulted in the increase to 98 species in the SMC. The analysis allowed the identification of 12 species that expanded their geographic distribution range, and 14 species were updated to a current scientific name. These results highlight the importance of natural protected areas in the preservation of diversity and abundance of species. The area with less anthropogenic disturbance "Punta Herrero" showed the highest number of species, contrasting with the low diversity in the southern locality of Mahahual, probably related to an increase in the residual waters and accumulation of plastic debris and Sargasso. Finally, these results contribute to the knowledge of the Caribbean crustaceans and are useful for adequate strategies for conservation and management of the regional fauna. It is important to continue the monitoring of the biological diversity of decapods in the Southern Mexican Caribbean, since they are an essential part of the ecosystems, and some of them are food resources appreciated by Mexican society.

**Supplementary Materials:** The following supporting information can be downloaded at: https://www.mdpi.com/article/10.3390/d14080649/s1, Table S1: List of current Achelata and Brachyura decapods recorded in the Southern Mexican Caribbean. Figure S1: Full BOLD Taxon ID Tree. References [51–79] are cited in Supplementary Materials.

**Author Contributions:** Conceptualization, J.J.-G., M.V.-M. and R.R.-L.; methodology, J.J.-G., M.V.-M. and R.R.-L.; Software, J.J.-G. and M.V.-M.; Resources, R.R.-L. and M.V.-M.; writing—original draft preparation, J.J.-G. Writing—Review & Editing, R.R.-L. and M.V.-M. All authors have read and agreed to the published version of the manuscript.

**Funding:** This research was funded by CONACyT with the project "Isotopic niches of key marine invertebrates to understanding the degradation of coral reefs in the Caribbean (ORDECYT-PRONACES/1312440/2020)**,** and the postdoctoral research grant "Estancias Posdoctorales por México 2020–2021" (J.J.-G.). The Mexican Barcode of Life (MEXBOL) node Chetumal, assisted with DNA extraction, PCR reactions, and sequence edition of all material presented here.

**Institutional Review Board Statement:** Not applicable.

**Informed Consent Statement:** Not applicable.

**Data Availability Statement:** Not applicable.

**Acknowledgments:** We thank Manuel Elías Gutiérrez and Alma Estrella García Morales (ECOSUR) for helping us in the processing of DNA Barcoding, as part of the CONACyT network Mexican Barcode of Life (MEXBOL). Special thanks are given to Nancy Cabanillas Terán (ECOSUR), Isabella Pérez Posada (ECOSUR), Víctor Conde Vela (UANL), Javier Nolasco Tinoco (I.T. de Chetumal) for helping us with the collection of biological material. Thanks to Sara Covarrubias (UMSNH) who elaborated on the localities map. This work was supported partially by the projects "Cephalopods and Crustaceans of the Mexican Caribbean Sea" and "Isotopic niches of key marine invertebrates to understanding the degradation of coral reefs in the Caribbean (ORDECYT-PRONACES/1312440/2020)". J.J.-G. would like to thank CONACyT for the postdoctoral research grant belonging to the program "Estancias Posdoctorales por México 2020–2021".

**Conflicts of Interest:** The authors declare no conflict of interest.

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
