# Peer review of "An Approach to the Diversity of Achelata and Brachyura (Crustacea, Decapoda) from the Southern Mexican Caribbean"

_diversity, doi:10.3390/d14080649_

Round 1

Reviewer 1 Report

General comments

 This manuscript provides results of a study into the biodiversity of lobsters (Achelata) and true crabs (Brachyura) of the Southern Mexican Caribbean, a poorly studied area. The authors used a mixed approach, reviewing the literature on the subject and collecting specimens in the study area to obtain morphological, ecological, and molecular data.

The manuscript is interesting and increases the knowledge on decapods of the study area, but some parts (e.g. 2.2 Material collected) can be improved, and the entire manuscript requires moderate to extensive English editing.

Specific comments

Line 10. Change “portunids” for “true crabs”. Portunids are just one type of Brachyura.

Line 28. Thalassinidea is an unaccepted name (check the World Register of Marine Species). I suggest removing this name and rather adding Achelata.

Lines 113- Suggest removing “randomly” and just leave “selected”.

Line 115- This section (2.2 Material collected) requires more detail. How were the decapods “grabbed” from algae and coralline rocks? Were they caught by hand? Or were the substrates (algae, coralline rocks) extracted and then searched for decapods (as implied further, in lines 117-118)?

Line 117- “The sampling effort to collect individuals was 5 hours per site”: Was this done using scuba? Or by free diving? Was it done by one person, two persons, or how many persons?

Line 134- Change “over” to “oven”

Line 157- Suggest removing parentheses.

Line 178- Change “Sargasso” to “Sargassum” and italicize.

Line 234- Change “Personnel” to “Personal”.

Lines 235-238. I suggest providing some background information (perhaps in the Introduction?) about the massive influxes of Sargassum occurring in the Caribbean over the last few years. Otherwise, the readers may not understand what these lines refer to.

Author Response

General comments

The manuscript is interesting and increases the knowledge on decapods of the study area, but some parts (e.g. 2.2 Material collected) can be improved, and the entire manuscript requires moderate to extensive English editing.

Response: We accept the comments to part " 2.2 Material collected". We also sent the manuscript to an external editor for English review.

Specific comments

Line 10. Change “portunids” for “true crabs”. Portunids are just one type of Brachyura.

 Response: Accepted and modified

Line 28. Thalassinidea is an unaccepted name (check the World Register of Marine Species). I suggest removing this name and rather adding Achelata.

Response: Accepted and modified

Lines 113- Suggest removing “randomly” and just leave “selected”.

Response: Accepted and modified

Line 115- This section (2.2 Material collected) requires more detail. How were the decapods “grabbed” from algae and coralline rocks? Were they caught by hand? Or were the substrates (algae, coralline rocks) extracted and then searched for decapods (as implied further, in lines 117-118)?

Response: We modify this part and include in the lines 115 to 118 the following text: "At each site, samples of decapods were collected manually during free diving samplings, mainly of coralline rocks associated with algae at depths not exceeding 3 m (collection permit PPF/DGOPA-060/20). The sampling effort was 5 hours by site and was performed by two persons. "

Line 117- “The sampling effort to collect individuals was 5 hours per site”: Was this done using scuba? Or by free diving? Was it done by one person, two persons, or how many persons?

Response: We accept the changes and the answer is included in the previous comment.

Line 134- Change “over” to “oven”

Response: Accepted and modified

Line 157- Suggest removing parentheses.

Response: Accepted and modified

Line 178- Change “Sargasso” to “Sargassum” and italicize.

Response: Accepted and modified

Line 234- Change “Personnel” to “Personal”.

Response: Accepted and modified

Lines 235-238. I suggest providing some background information (perhaps in the Introduction?) about the massive influxes of Sargassum occurring in the Caribbean over the last few years. Otherwise, the readers may not understand what these lines refer to.

Response: We modify this part and include in the line 86 the following text: "This knowledge is essential to make sustainable use of natural resources, since there are currently various threats to public health, ecosystem health, fishing and the economic development of the region, such as the growing arrival of Sargassum in the Mexican Caribbean, which generates hypoxia and deterioration of water quality, affecting individuals of a large number of species, mainly fish and crustaceans [31]."

Reviewer 2 Report

The authors presented an interesting manuscript on the biodiversity of Decapoda in the Southern Mexican Caribbean. I read the manuscript with great pleasure. The presented work and results constitute a considerable contribution to the diversity and distribution of both studied decapod groups (Achelata and Brachyura) in the region. In other words, this work will be of special importance for future biodiversity and ecological surveys, constituting a solid base for further research. The authors should be especially priced for extending the checklist of the taxa occurring in this region by adding six new records and for correcting and updating binominal formal names for 14 different decapod nomina. The authors conducted a genetic study in which the genetic diversity they have discovered could be assigned to 21 taxa, of which one could be a potential new for science species. I really hope that the authors will dig deeper into this issue which will result in a correct final identification with already known nominal species or a formal description of species new for science. I do not have any specific comments or corrections for the MS, and I think it could be accepted in its current stage. My only concern during the reading was on lines 69-71, where the authors mention a range 3.7% and 4.9% to be a normal interspecific genetic divergence in crustacean decapods. I was wandering if they really meant this or if they were thinking that the possible arbitrary threshold for genetic species delimitation in this group can be placed somewhere in this range. So my only small recommendation is to clarify this sentence.

Author Response

Comments and Suggestions for Authors

My only concern during the reading was on lines 69-71, where the authors mention a range 3.7% and 4.9% to be a normal interspecific genetic divergence in crustacean decapods. I was wandering if they really meant this or if they were thinking that the possible arbitrary threshold for genetic species delimitation in this group can be placed somewhere in this range. So my only small recommendation is to clarify this sentence.

Response:  We accept the comment and believe that this range of values is an arbitrary threshold that allows genetically delimiting different species of decapod crustaceans.